# Change the Humans First: Principles for Improving the Management of Free-Roaming Cats

**DOI:** 10.3390/ani9080555

**Published:** 2019-08-14

**Authors:** Lynette J. McLeod, Donald W. Hine, Aaron B. Driver

**Affiliations:** 1School of Psychology, University of New England, Armidale, NSW 2350, Australia; 2UNE Business School, University of New England, Armidale, NSW 2350, Australia

**Keywords:** human behaviour change, intervention design, behaviour change wheel, community-based social marketing

## Abstract

**Simple Summary:**

For free-roaming cat management to be effective, people—including land managers, conservationists, cat lovers and the general public—need to be sufficiently empowered and motivated to accept and implement management actions. Research in the social and behavioural sciences has shown that engaging everyone and gaining consensus can be incredibly challenging. This paper describes an integrative framework based on the behavioural literature to design better, equitable and ethically acceptable interventions for free-roaming cat management.

**Abstract:**

In Australia, free-roaming cats can be found in urban and rural areas across the country. They are inherently difficult to manage but it is frequently human behaviour that demands the most attention and is in most need of change. To the frustration of policy makers and practitioners, scientific knowledge, technological developments, and legal and institutional innovations, often run afoul of insufficient public capacity, opportunity and motivation to act. This paper demonstrates how the behavioural science literature can provide important insights into maximising the impact of free-roaming cat control activities within an ethical framework that prioritises acting “with” all stakeholders, rather than “on” stakeholders. By better understanding how human values, attitudes and beliefs are shaped, practitioners can more effectively and respectfully interact with how people interpret the world around them, make choices and behave. This literature also has much to say about why certain types of media and marketing messages elicit behaviour change and why other types fall flat. Finally, in addition to explaining the behavioural science and its implications, this review provides researchers, policy makers and engagement specialists with an inclusive, practical framework for conceptualising behaviour change and working to ensure land managers, cat owners and the general public can agree on and adopt best practices for managing free-roaming cats.

## 1. Introduction

The domestic cat, *Felis catus*, can be found in urban and rural areas across the globe. It has the uncommon distinction of being a valued companion animal and one of the world’s worst invasive mammal species [1]. Cat ownership and companionship offer many benefits (e.g., [2,3]). However, mounting evidence indicates that free-roaming cats create substantial negative impacts through predation and competition, threatening the existence of many wildlife species in Australia and worldwide [4,5,6,7]. Cats can also transmit diseases to humans, native animals and livestock, either directly [8] or through faecal contamination of pastures and waterways [9,10]. Any free-roaming cat, regardless of its location or dependence on humans, has the potential to cause harm [11,12], as well as facing many hazards itself, including threats from vehicles, people, other animals and cat-specific diseases [13,14,15].

The management of free-roaming cats is complicated by the dual nature of this animal. For example, in Australia, cat management falls under two distinct policy frameworks: companion animal ownership and environmental pest management. Cats are designated into arbitrary groups based on how and where they live in order to distinguish which individuals are covered by which legislation, i.e., between the valued companion animals and wild, free-living individuals. To further complicate the situation, the terminology and definitions of these groups are not consistent in Australia, nor globally, and are used interchangeably. Common distinctions include owned/unowned, domestic/non-domestic, outdoor, semi-owned, stray, wild and feral [16,17,18].

Companion animal legislation is usually managed by local jurisdictions, and encourages—and sometimes enforces—‘responsible’ cat ownership. This may include de-sexing to prevent unwanted breeding, registration and identification (e.g., microchips and collars) to assist in ownership issues, as well as some form of containment (e.g., keeping cats within the owner’s house and yard either at night or all the time) to reduce predation, disease and injury risks, or designating cat-free areas to reduce predation impact on local wildlife [19,20,21]. Some jurisdictions may also encourage community members to act responsibly towards cats that are living freely—that is uncared for strays, ferals and cat colony members. Strategies include discouraging people from feeding these animals to instead, in some instances, taking responsibility by arranging de-sexing and health care to improve the animal’s overall welfare and reduce the harm they cause [18,22,23,24]. When adoption strategies are not feasible, people are often encouraged to report unowned animals to the appropriate authorities so that they can be looked after by someone else responsibly, rehomed or removed [25,26].

Australia is unique in that it has declared the ‘feral’ cat a national pest, and listed predation by feral cats as a key threatening process at a national legislative level. This declaration and listing under the Commonwealth *Environment Protection and Biodiversity Conservation* Act 1999 has given priority to feral cat management in threatened species recovery programs. It has also allowed the development of a national framework to guide and coordinate research, management and other actions needed to ensure the long-term survival of those native species and ecological communities affected by feral cat predation (https://www.environment.gov.au/biodiversity/invasive-species/feral-animals-australia/feral-cats). Both lethal and non-lethal methods are used to manage free-roaming cats. Lethal methods include poison baits, trapping and shooting. Non-lethal methods include exclusion fencing, habitat management using fire, grazing, and limiting access to potential resources such as food and shelter (for a review of control methods, see [27,28]). One of the major components of any of these programs is to gain and maintain a ‘Social Licence’ to operate, i.e., establish and maintain social legitimacy, credibility and trust of the community to gain the acceptance and approval of any management activities. This is usually done by engaging and building relationships with community stakeholders, involving them to some degree in the decision making, defining and negotiating agreements to manage expectations and improve transparency, and creating opportunities to work together and generate shared experiences [29,30]. This can be a challenging exercise as many diverse, and sometimes seemingly opposing, views and values exist towards free-roaming cats, their impact on the environment, and the ethics of managing this species including humaneness of control methods [31,32,33]. These aspects are discussed in more detail in the following section.

## 2. Issues around Designing Interventions for Human Behaviour Change

For free-roaming cat management to be effective, people—including land managers, conservationists, cat lovers and the general public—need to be sufficiently empowered and motivated to accept and/or implement management actions, as well as be dissuaded from engaging in behaviours that undermine management outcomes [34]. Research in the social and behavioural sciences has shown that this can be incredibly challenging and requires a sophisticated understanding of human behaviour and its underlying determinants—the internal and external factors that increase the likelihood of desirable behaviours and reduce the likelihood of undesirable ones. It also requires a detailed knowledge about how behaviour changes over time, and which intervention types are most effective in driving behaviour change in which contexts [35].

In addition to understanding the theoretical and pragmatic aspects of behaviour change, policy makers and practitioners must carefully consider the ethical implications of their interventions [35,36,37,38]. One common criticism of behaviour change interventions is that they involve “acting on” people (often without the consent or conscious awareness of targeted individuals). Government strategies to control and change personal behaviour raise significant philosophical questions about boundaries between the private and the public, the limits of state control, the ability of individuals to act rationally and independently, and fairness to all individuals [35,38]. We agree that these are all important issues. But it is equally important to appreciate that nothing is inherent in the behaviour change theory or methodology that directly advocates or necessitates ‘top–down’ command and control. Indeed, a common approach adopted by behaviour change practitioners is to identify and engage with all stakeholders who are involved in, and who will be potentially impacted, at all stages of the strategy development process. That is, the aim should be to work with, not on, individuals, organisations and communities. This inclusive approach to behaviour change is particularly important for ‘wicked problems’ such as free-roaming cat management, which are not only challenging from an ecological control perspective, but also involve a complex mix of stakeholders with divergent values and preferred solutions [35,39,40]. We acknowledge that this consultation can be messy (for example, funding cat ‘sanctuaries’ where free-roaming cats can live out their life in a healthy, safe environment may be a solution for some stakeholders but a non-viable option for others [41]), but also that there are proven strategies for systematically building consensus when dealing with diverse stakeholders [42]. We discuss this important issue of stakeholder consultation and engagement in more detail in the sections that follow.

## 3. Understanding Human Behaviour

Research into human behaviour is extensive, with many theories describing factors that exert a causal influence on behaviour as well as the nature of this influence (for example Michie, West [43] describe 83 theories of behaviour relevant for health issues alone). Theories of behaviour change also describe the process of change, and how people learn and change over time [35,43]. These theories of behaviour and behaviour change are useful for identifying the main drivers of behaviour, and the internal and external barriers that sometimes prevent behaviour change.

Consequentialist models of behaviour assume that behaviour arises from a deliberate decision-making process involving a systematic evaluation of potential costs and benefits associated with a range of behavioural options, i.e., rational decision making. They view conscious expectancies about future outcomes as the key driver of decision making. The theory of planned behaviour (TPB) [44] is perhaps the most broadly applied consequentialist theory of human behaviour. According to TPB, the primary determinant of cat management behaviours is an individual’s conscious decision—or reasoned intention—to engage in one or more of these cat management behaviours. In turn, TPB proposes that intentions are determined by three main psychological factors: (1) attitudes (the extent to which we feel positive or negative towards the behaviour), (2) subjective norms (the extent to which important others in our lives think that engaging in the behaviour is a good idea), and (3) perceived behavioural control (the extent to which we believe we can successfully engage in the behaviour). Thus, according to the theory, if people have positive feelings about a particular cat management action, expect they will receive social approval for doing it, and believe they have the knowledge, skills and resources to complete relevant actions, then they will be more likely to develop intentions to act and initiate the behaviour.

A range of studies has shown one or more TPB variables to be important predictors of intentions and/or behaviours related to responsible cat ownership and feeding free-roaming/stray cats [18,20,45,46,47,48]. However, as Khor et al. illustrated with their incorporation of ‘anticipated regret’ [38], TPB alone offers little insight into other important social and intrapersonal factors. Psychological research has identified a range of other factors encompassing emotions, role beliefs, personal norms (morals), and habits that may also be important determinants of behaviour (see the following reviews: [34,35,43]).

Although behaviour models allow us to understand specific behaviours, they do not necessarily indicate how behaviours can be changed. Thus, a complementary body of literature on behaviour change also exists, based not only on social psychological processes, but drawing on research from a diverse range of disciplines such as education theory, cybernetics and engineering. These models incorporate features such as:Temporal processes that describe behaviour and change over time, for example the Transtheoretical Stages of Change Model [49];How people learn, which is considered to be fundamental to the process of change, for example Self-Regulated Learning [50];Systems thinking, which looks at interactions of the whole [51,52];Social change through theories of diffusion and social capital, for example Rogers Diffusion of Innovations [53], which describes the process of adoption of innovation using interactions within social networks.

The challenge for practitioners lies in knowing which model to choose, and when, from this potentially overwhelming ‘universe’. To assist practitioners, a number of frameworks have been created, offering a methodical approach to develop more effective interventions without having to grapple with the vast behavioural change literature. For example, in health, Michie, Atkins [36] created the Behavioural Change Wheel (BCW), McKenzie-Mohr [54] developed Community-Based Social Marketing (CBSM) to encourage pro-environmental behaviours, and Mindspace has been used to help in government policy areas [55]. Although different in structure and terminology, these frameworks in most part are built around four guiding principles:Focus on human behaviour—identify and select behaviours that will produce effective outcomes;Know your audience—understand the causes of behaviour and variation in causes across target populations;Match interventions to the primary cause of behaviour;Use science-based evaluation to determine what works and why.

Given that no specific behaviour change framework exists for animal management, in this paper, we will describe how an integrated framework can be used to develop effective behaviour change interventions to improve free-roaming cat management outcomes. We begin by demonstrating a process for selecting which behaviours to target. We then describe how to identify the main causes of desired behaviours—both the drivers that increase the likelihood the behaviours will occur and the barriers that prevent them. Then, we show how to match interventions to the primary causes of behaviour and conclude by discussing the importance of determining what works and why, using science-based evaluations.

## 4. Development of Interventions

### 4.1. Principle 1: Focus on Human Behaviour

Before we can understand the role of human behaviour in improving cat management, it is important to be clear about the particular problem that we are trying to solve. Identifying and engaging with all stakeholders at the beginning of the process is critical to gain a better understanding of people’s perception of the nature of the problem, including their understanding of possible causes and possible solutions. It is also important for behaviour change practitioners to understand how different stakeholders are being affected, their priorities and interests, as well as potential conflicts between personal interests and what is collectively desirable [36,37]. For example, if a local authority is receiving a growing number of complaints from community members regarding ‘nuisance’ cats (both owned and unowned), prior to developing an intervention strategy to address the problem, the authority should consult key stakeholders. These stakeholders might include cat owners, affected community members, local vets, shelter operators, biodiversity scientists, animal ecologists and council staff. After consulting the main stakeholders, four collectively determined objectives might emerge:Improvement in the welfare of these cats;A reduction in the number of these cats;Protection of wildlife and environmental assets from predation by these cats;Protection of wildlife, livestock and humans from diseases and parasites that may be transmitted by these cats.

With the objectives defined, it is now possible to consider what types of human behaviours contribute to the problem, what types of behaviours can resolve the problem, and who is involved in these behaviours. One useful approach is to list the key behaviours needed to achieve this goal. Table 1 illustrates this process by listing the candidate target behaviours according to who (for simplicity it is just illustrating the owned cat aspect of this issue, therefore cat owners) needs to do what, when, where, and how often [36].

In most cases, interventions should aim to influence a small number of highly positive impact behaviours, which have a high probability of being adopted, and are currently not widely practiced within the target population [54,57]. The local authority in our example does not want to spend time, energy and money convincing people to engage in activities that will have little positive impact on their free-roaming cat problem. Nor do they want to waste resources trying to influence behaviours that are unlikely to be adopted, or that everyone is already performing. In his book on community based social marketing, McKenzie-Mohr [54] proposed a simple framework for prioritising behaviours (the Behaviour Prioritisation Matrix—BPM) based on the:Effectiveness of the behaviour in bringing about positive outcomes;Likelihood that the behaviour will actually be adopted;Proportion of the target population already engaged in the behaviour (penetration).

The behaviours are then ranked based on the followingequation:*Total Weighted Impact* = *Effectiveness × (Maximum Possible Penetration* − *Observed Penetration) × Likelihood of Adoption*

Table 2 illustrates this process for the key behaviours that were listed in Table 1. In this instance, although de-sexing has the highest impact on reducing the number of free-roaming cats, it already has a high penetration (93% of the population surveyed claimed that their cats were already de-sexed), thus making it less desirable for a local authority than the higher ranked behaviour—24-h containment. This latter behaviour also has a relatively high impact, but the current low penetration and moderate likelihood of adoption may result in a bigger ‘bang’ for a limited budget. This result varies from that of Linklater, Farnworth [21], who found, in their context, that a night curfew was the behaviour most suited for their campaign.

Kneebone et al. [57] offer an alternative method to identify and prioritise target behaviours, using a visual, Impact–Likelihood matrix. This matrix is created by mapping the effectiveness of a behaviour (or impact) and the likelihood of adoption on a grid, overlaid with data on current participation. Priority behaviours can then be identified by their location within the grid, while retaining other useful information, such as how the behaviours relate to one another. Figure 1 illustrates this process for the same data that is presented in Table 2.

A behaviour with high impact and a high likelihood of adoption (e.g., de-sexing) may have first priority to be chosen because it is relatively easy to adopt and has a large impact on the issue. However, as in this case, these behaviours may already have a high adoption rate (i.e., there is only a small proportion of the target audience who have not already adopted this behaviour as represented by the smaller circle in Figure 1), so it may not be the best behaviour to target. Behaviours that have high impact but a lower likelihood of adoption may provide better potential targets, because they tend to have low current penetration, but will require more work and resources to be adopted (e.g., 24-h containment). Behaviours falling in the low impact, high likelihood of adoption lack effective impact on the issue but because they are easy to adopt, they might act as a catalyst to encourage more difficult behaviours in the future [61]. For example, getting people to initially introduce a cat curfew may be the first step to adopting 24-h containment (McLeod unpublished data). Behaviours with a low impact and low likelihood of adoption (e.g., registration) are low priority, because they achieve little to address the issue and are hard to adopt [57].

### 4.2. Principle 2: Know Your Audience

To design the most effective behaviour change strategies, it is necessary to collect information on the actual barriers that impede engagement in the desired behaviour as well as those factors that drive action. It is important to collect broad data on these possible causes, including internal factors such as values, beliefs and knowledge, and external factors such as community norms and physical opportunities of performing particular behaviours. Possible sources include research literature, expert and stakeholder knowledge and, most importantly, information from the target audience (either from direct observations, if appropriate, or using quantitative and qualitative survey methods such as questionnaires, interviews and focus groups) [62,63]. Relying on just one source may overlook critical factors that drive management activities or prevent these activities from occurring. For example, just relying on expert evaluations may miss important insights on individual’s beliefs, emotions or capabilities that may not be evident to an external observer [21,64]. Once this information has been collected, statistical modelling can be used to determine which of these factors are most useful in discriminating between those who engage in the target behaviour and those who do not [48,65].

In a comprehensive review of the literature, Michie et al. [43] identified 83 psychological theories of behaviour, each incorporating multiple causes factors. This suggests that there can be a very large number of potential drivers and barriers relevant to understanding any particular behaviour. McLeod et al. [34] demonstrated that most drivers and barriers relevant to invasive animal management can be classified into the three categories described by Michie and her colleagues in their overarching, integrative model of behaviour, the Capability, Opportunity, Motivation-Behaviour (COM-B) model [66]. This COM-B model can help practitioners understand behaviour in context by identifying the main factors that influence the adoption of desired behaviours, for example de-sexing pet cats. It also helps identify what exactly needs to change to increase the likelihood that these desired behaviours will occur.

According to the model, behaviour is determined by three main factors:*Capability*. An individual’s physical and psychological capacity to engage in the behaviour of interest. COM-B distinguishes between two types of capability. *Physical capability* refers to the extent to which an individual can engage in the behaviour. For example, does an individual have the physical ability to install a cat-proof fence? *Psychological capability* refers to the capacity to engage in the necessary mental activities (risk assessments, mental simulation of possible outcomes, decision making, etc.) to select and implement an appropriate course of action.*Opportunity*. These are factors external to the individual that prompt or enable the behaviour to occur. COM-B distinguishes between two types of opportunity. *Physical opportunity* refers to situational factors such as having relevant equipment or supplies readily available that are needed to address the problem. It can be difficult to set a trap if a trap, or the resources to build a trap, are not available. *Social opportunity* refers to cultural or community values and norms that may make engaging in recommended best practices more or less likely. For example, if most cat owners within a neighbourhood are keeping their cats in at night, this creates a social norm that increases that likelihood that others in the neighbourhood will also engage in this practice.*Motivation*. These are factors internal to the individual that energise or direct behaviour. COM-B distinguishes between two types of motivational factors [67]. *Reflective motivation* consists of conscious deliberation and reasoning, and often involves evaluating threats, planning, goal setting, and mentally simulating possible outcomes associated with various types of actions. For example, prior to deciding to control feral cats, a land manager may make a list of the costs and benefits of engaging and not engaging in the various control activities and select the option that he or she believes is most likely to produce the most positive outcome. *Automatic motivation* refers to mental processes that operate largely outside conscious control of the individual, including habits, impulses and emotionally driven behaviour. For example, a cat owner’s ‘decision’ to keep their cat contained may be emotionally based after witnessing the injuries suffered by their cat after being hit by a car.

The pattern of drivers and barriers influencing a given behaviour may vary across individuals within a community. Not everyone views free-roaming cats, their impacts and their management in the same way. Distinct segments of the community may have very different driver-barrier profiles, reflecting their values, beliefs and current behaviours. Thus, a practitioner may not be dealing with a single target community, but rather several, and may need to craft and refine interventions with the intended target audience in mind.

Understanding these audience segments can assist engagement practitioners to make four strategic decisions [68,69]:*How to best optimise interventions for each audience using their unique COM-B profiles*. People who do not have the capability to perform a desired behaviour will require a different intervention to those who are capable but need the motivation to act.*How to determine who should be targeted*. To maximise on-the-ground impact it may be better initially to target a large group of disengaged but receptive audience members, rather than focusing on a smaller group who are not interested, and who would require more time and money to engage.*How to ensure the audiences engage with the intervention*. Different audience segments may have their own unique preferences for where they obtain information, with some using social media such as Facebook, others preferring traditional print media, and others relying on face-to-face interactions.How to select the best messengers with the relevant expertise, values and personal experiences needed to build and maintain trust with their audiences. Not all audiences will perceive certain communicators as credible and trustworthy.

### 4.3. Principle 3: Match Interventions to the Primary Causes of Behaviour

Any intervention may rely on one or many behaviour change techniques to achieve its outcomes, but not all techniques are equally well suited to all situations. For example, if an identified barrier to de-sexing pet cats was that owners did not have access to a veterinary practice in their local community, then offering reduced rates for the operation alone may not be the best way to improve uptake. The efficiency and impact of interventions can only be improved by ensuring that the techniques selected complement the types of driver and barrier factors identified. A better option may be to bring a visiting vet to the local community or provide transport to the closest vet practice.

Linking specific driver and barrier factors to suitable behaviour change techniques can be challenging. The advantage of the COM-B model is that it allows practitioners to link the identified COM-B mechanisms that drive and impede the desired behaviours, i.e., Capabilities, Opportunities and Motivations, to the most appropriate behaviour change technique (Table 3). As part of their BCW model, Michie and her colleagues identified nine general behaviour change techniques: education, training, persuasion, incentivisation, coercion, restriction, environmental restructuring, modelling, and enablement, each of which is further linked to seven discrete policy tools (plans of action and strategies to help governments and organisations achieve their outcomes): communication/marketing, guidelines, legislation, regulation, fiscal, environmental/social planning and service provision [36,66].

To address the appropriateness and feasibility of any potential intervention design Michie and her colleagues developed the APEASE system which uses six criteria [36]:Affordability—can the intervention be delivered within an acceptable budget?Practicality—can the intervention be delivered effectively as designed in a real-world context?Effectiveness—what impact will the intervention have in relation to the desired outcomes in a real-world context? And is the expected impact worth the cost required to achieve it?Acceptability—is the intervention judged to be appropriate by key stakeholders?Side-effects—are there likely any unwanted side-effects or unintended consequences? These are sometimes difficult to predict, but definitely worth considering.Equity—does the intervention produce disparities between different sectors of society?

Crucial to the success of any intervention is a good communication strategy. It not only creates awareness about the intervention but itself may be part of the behaviour change technique, i.e., persuasion. There are two main components of a communication strategy: content—the substantive material covered in the communication (i.e., the subject, ideas or messages), and its delivery—the means of presenting and distributing this content to the target audience (e.g., mass media broadcast, social media, printed material, word of mouth). While there has been rapid development and innovation in communication devices and distribution channels in recent decades, well-conceived content remains at the core of an effective communication strategy [70]. It should be specific and communicate key ideas, using appropriate language and structure, while not being open to any adverse interpretation.

Selecting the appropriate message framing, i.e., how the issue is presented, is an important consideration [71,72]. Many target audiences of free-roaming cat communications have strongly held beliefs and attitudes on the subject [41,73,74,75,76,77]. Using frames that match audience values and concerns will get the message noticed, processed and acted upon [72,78]. For example, a large number of campaigns that encourage cat owners to contain their pets have used messages framed around saving wildlife to motivate cat owner action. However, recent research suggests that framing around the benefits to cats’ welfare may be more effective in engaging some cat owners [24,47,79].

An important aspect of free-roaming cat communication may be to debunk any misinformation [80]. Debunking misinformation requires more than just giving the right information. In a review of online free-ranging cat communications, McLeod, Driver [81] found that over three-quarters of the cases that attempted to debunk misinformation may have inadvertently reinforced the myths by mentioning them before attempting to correct the misinformation with the right information. Cook and Lewandowsky [82] refer to this as the ‘familiarity backfire effect’ where the myth is repeated first, thus making it more familiar and more likely to be accepted. To prevent this problem, they suggest avoiding mentioning the myth entirely, or when this is not plausible, lead with a core fact immediately followed by strong, clearly stated evidence before acknowledging the misinformation. For example, instead of starting with a statement such as, “Many people believe that pet cats need to roam to be happy”, thereby instilling the idea that pet cats may need to roam to be happy, a more preferable opening statement might be that “a happy pet cat is one that is kept safe and entertained at home”.

To select an appropriate delivery mechanism, it is important to consider the strength of the identified drivers and barriers, how predisposed an individual is to change, message type and how it will spread, as well as the resources at your disposal [36,54,83,84,85]. Mass media channels (one individual or entity transmitting to an audience of many) are usually the most rapid and efficient means of promoting awareness and changing attitudes (e.g., [36,86]). Examples of this include the many television and social media advertising campaigns for welfare or cat advocacy organisations. On the other hand, interpersonal channels, such as face-to-face exchange between two or more individuals, are more effective in promoting actual behavioural change (e.g., [36,87]). Hence adoption agencies taking the time to talk to potential cat adopters, or a cat management officer from a local authority talking to a small group of local cat owners, may be a more effective method to encourage ‘responsible’ cat ownership behaviours (e.g., The Cat Protection Society of New South Wale’s ‘Good Neighbours Project’: https://catprotection.org.au/responsible-cat-ownership/).

### 4.4. Principle 4: Use a Science-Based Evaluation to Determine What Works and Why

Each year, substantial investment is made in developing interventions to engage the public about free-roaming cat management issues. But too often no scientifically credible evidence about the effectiveness of these interventions is collected [88,89]. These types of unsatisfactory outcomes are entirely preventable by ensuring that an appropriate evaluation plan is tied to each intervention. The design of an effective evaluation plan should be guided by ‘best-practice experimental design’ principles [90,91]:*Always include a control group*. A minimum requirement is the inclusion of a control group, i.e., a group of people that does not experience the intervention. Without a control group, it is impossible to know whether a change in behaviour or outcomes is due to the intervention or an infinite number of other uncontrolled factors, such as an increase in public interest driven by media reports, or an overall increase in the targeted issue.*Whenever possible, use random assignment*. Randomised controlled trials represent the gold standard for evaluating a treatment or project across many scientific disciplines. Random allocation to experimental conditions removes any bias and allows resulting observed differences between the treatment and control groups to be attributable to the intervention, and not pre-existing group differences or other uncontrolled factors.*When random assignment is not possible, use quasi-experimental designs*. Unfortunately, working with people in ‘real-world’ situations does not always allow for random assignment of specific treatment groups. Quasi-experimental design, which compares naturally occurring or self-selected groups, is the next preferred option [91]. For example, if you launch an intervention promoting the reporting of free-roaming cats in a community, you could compare the reporting practices between those who are aware of the intervention and those who were not. Compared to a randomised control experiment, quasi-experimental designs will not give you the same level of confidence that your intervention was the main factor driving the behaviour change.*Use statistical tests to evaluate effects*. The use of statistical tests increases the certainty that the measured differences between treatments are ‘real’ and meaningful, and not simply due to chance. For example, what if you found that your intervention resulted in a 10% increase in participation but, over the same time, there was a 5% increase in the control group? Has your intervention produced a meaningful difference, or was it simply due to chance variation? Without statistical tests there would be no reliable way of knowing. If you are unfamiliar with the many tests and software options, consultations with a qualified statistician is a worthwhile endeavour.*Measure actual behaviour change*. In any intervention evaluation plan, it is important to measure actual behaviour change. Showing that an intervention has increased awareness, or changed attitudes, or even the intention to act, can all be important discoveries. But changes in awareness, attitudes and intentions do not always translate into behaviour change [92]. Thus, behaviours need to be measured directly. This may not be an easy thing to do when working with people in ‘real-world’ situations, but, where possible, direct observations of desired behaviours are preferable over self-reported behaviour. People’s actions do not always match their claims [93].*Link behaviour change to on-ground impacts*. Another important aspect of an evaluation plan is to link behaviour change back to the on-ground impacts and management outcomes. It is often assumed that getting people to adopt recommended management practices or new technologies will automatically deliver positive outcomes such as increased biodiversity. These assumptions should be assessed. If an expected increase in local wildlife following a cat management program does not occur, the original assumption linking cat predation to wildlife survival may be incorrect. Or perhaps other unanticipated factors, such as environmental degradation or fox predation, may be at play. These need to be investigated and clarified.*Evaluate the cost-effectiveness of interventions*. Resources are almost always limiting in any cat management program. So not only is it important to measure success with behaviour change and outcome achievement, it is also important to calculate the cost-effectiveness, that is, the overall benefit per dollar spent. Cost-effectiveness calculations can help you choose between competing strategies. If two strategies are as effective as one another, but one costs substantially less, it makes sense to select the cheaper option for future implementation.

## 5. Summary and Conclusions

We have shown how behavioural science approaches can be used to improve the effectiveness of interventions to change human behaviour and potentially improve free-roaming cat management programs. To achieve this, we have demonstrated an integrative approach, selecting the components that are most relevant and useful in the cat management context, incorporating the following key features:Communication and consultation with all stakeholders at all stages of the process is essential to achieving effective, equitable and ethically acceptable outcomes.An emphasis on defining the issue in behavioural terms and understanding the relevant behaviours is needed first. The design process should not start with a strategy and work backwards.Determinants of target audience behaviours should be identified from a range of sources including direct observations, interviews, surveys and focus groups.Because many free-roaming cat management programs require collective support and action from a range of community members from various backgrounds, with varying capabilities, opportunities and motivations, audience segmentation (the identification of audience subgroups that have similar values, interests, needs and behaviour patterns) is a beneficial technique to apply.Chosen intervention strategies must match the primary causes of behaviour. Do not be tempted to implement something following the “it seemed like a good idea at the time” principle [47].Interventions need to be evaluated. Changes in target audience behaviour, not just outputs, should be assessed before and after implementation, and ‘treatment’ groups should be compared to a control group not exposed to the intervention. The sharing of these results would accelerate learning about best methods for engaging and motivating those involved in free-roaming cat management.

Employing this knowledge of human behaviour change to free-roaming cat management programs has the potential to improve approval and participation rates, thereby improving the outcomes of future free-roaming cat management programs. However, it is important to acknowledge that applying this knowledge is no ‘quick fix’; it requires substantial effort to organise and evaluate potential influence factors and understand how these factors vary across contexts. These realities should be recognised by managers and funding bodies and built into any future free-roaming cat management programs.

## Figures and Tables

**Figure 1 animals-09-00555-f001:**
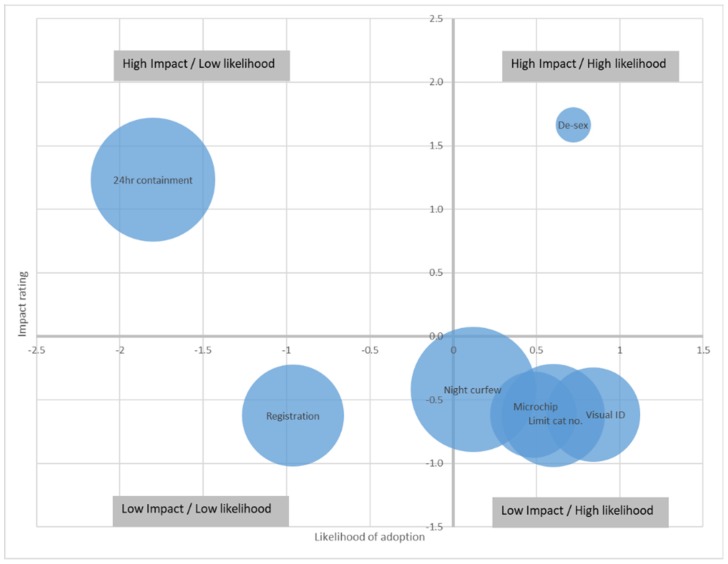
Impact–Likelihood prioritisation matrix ranking cat owner behaviours based on standardised values for effectiveness of behaviour for reducing the number of free-roaming cats (impact) and likelihood of behaviour adoption (after [56]). Size of circles indicate proportion of target population not currently engaging in the behaviour.

**Table 1 animals-09-00555-t001:** Listing key cat owner behaviours based on literature [19,20,21] and consultation with professionals from organisations involved in cat management [56].

What	Who	When	Where	How Often
De-sexing	cat owner	as early as possible	at local vet	once only
Microchipping	cat owner	when first adopted/purchased	at local vet	once only
Visual identification	cat owner	as early as possible	placed on cat	all the time cat the is outside
Registration	cat owner	when first adopted/purchased	at office/online	every year/or when details change
Night curfew	cat owner	at night	at home	every night
24-h containment	cat owner	all the time	at home	all the time
Limit no. of cats per household	cat owner	all the time	at home	all the time

**Table 2 animals-09-00555-t002:** Ranking of cat owner behaviours based on effectiveness of behaviour for reducing the number of free-roaming cats, likelihood of adoption by cat owners and current practice (penetration) (after [54]). Data for effectiveness from [56], and survey data for Likelihood and Penetration from Myriad Research [58], except where indicated.

Behaviour	Effectiveness	Likelihood of Adoption (0–4)	Penetration (0–1)	Weighted Score	Rank
24 h contain	8.10	1.4	0.13	9.87	1
Night curfew	1.60	3.0	0.12	4.22	2
De-sex	9.80	3.5	0.93	2.40	3
Visual identification	0.82	3.6 ^1^	0.50	1.48	4
Limit cat no.	0.79	3.4	0.40	1.42	5
Microchip	0.82	3.3	0.58	1.14	6
Registration	0.79	2.1	0.41 ^2^	0.98	7

**^1^** Murray, Sciggins [59], **^2^** McMurray [60].

**Table 3 animals-09-00555-t003:** Examples of intervention techniques for cat management and the Capability, Opportunity, Motivation-Behaviour (COM-B) behavioural factors they best match (after [66]).

Technique	COM-B Factor	Examples
Persuasion	Motivation	Messages that promote the benefits of a particular cat management behaviour or the negative implications of not performing the behaviour
		Provision of information so that comparison can be made for participating or not participating
		Messages framed in local context and delivered by locals
Incentives	Motivation	Positive financial or social reward for participating in a particular action, e.g., reduced registration for de-sexed cats
Coercion	Motivation	Fines or social punishment for not participating in desired behaviour, or participating in a non-desired behaviour
Restriction	Opportunity	Regulations to restrict a behaviour’s performance, e.g., limiting the number of cats a household can have
Modelling	Motivation	Setting up a ‘demonstration site’ at a well-known location, or with an inspirational local, e.g., to display how they contain their cat
Enablement	Capability Opportunity Motivation	Ensuring local supply of resources such as traps that can be loaned to so they do not have to be purchased, and conducting training in how to use them proficiently
Environmental re-structuring	Opportunity	Promoting a particular species as a pest in an area to highlight it as a social problem in need of a solution
	Motivation	Increasing access to services by offering more locations or extended opening hours
Education	Capability Motivation	Producing written material or video clips to disseminate and illustrate information on cat management techniques
Training	Capability	Running a workshop to provide practical and technical instructions on how to build a cat-exclusion fence

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
