# Peer review of "Change the Humans First: Principles for Improving the Management of Free-Roaming Cats"

_animals, 2019, doi:10.3390/ani9080555_

Round 1

Reviewer 1 Report

Reviewer comments

To the authors,

I have prepared extensive comments as I take my role as reviewer seriously. No abruptness is intended but I’ve tried to be short for the sake of brevity.

I found this a challenging paper to read. I do believe that animal welfare, conservation biology and human behaviour change are inextricably linked.

But I think there is a lack of clarity here and the paper needs to be revisited, restructured and rewritten in parts.

Ethically, I do question the position on manipulating those who have different views about cat management to the point where they will participate in such programs. I feel very uncomfortable about that. When I saw the title I hoped changing humans referred to changing behaviours like habitat destruction and fragmentation, pollution, pest control (ie which can decimate native rodent species) etc that free-roaming cats often “mop up”.

Specific comments below:

Line 12: “getting everyone on the same page” seems a bit of a leap and in an issue as divisive as free roaming cat management, highly unlikely despite the most exceptional communication. I wonder if an alternate expression would be better? This is repeated in lines 86-87 and I feel is not transparent. The authors acknowledge strong divergent opinions, yet want everyone on the “same page”, ie agreeing with a particular view. I think the authors are obliged to state their view up front.

Line 22: I would question the goal of maximising the impact of free-roaming cat activities. This is loaded. Isn’t the desired outcome maintenance or increase of biodiversity?

Line 35: Would this not also be the case for rabbits?

Line 72: It would be useful for the reader to cite these examples so they can read further.

Line 73: delete “control”. You could just use methods as killing is the form of population control utilised. There is no reference to whether these methods are humane or not. I think it is important to make this clear.

Lines 77-80: Surely social license is also predicated on the humaneness of killing and other control methods?

Line 83-86: There is also literature documenting that people are more likely to act if they believe their actions will make a difference. This is documented in a range of settings, whether its about using PPE to reduce the spread of equine influenza or implementing changes to reduce anthropogenic climate change.

Line 87: Whenever you state that research has shown something, you are required to cite that research. This paragraph makes strong statements but doesn’t tell us where this info is sourced from.

Line 94: Avoid stating “many theories”. How many? Also in line 99. Give an example, but don’t say “many”.

Line 105-106: What do you mean by “these behaviours” – which ones?

Line 140: Suggest add “we propose” re the common 4 principles

Line 160: Not all of these questions are answered by your points. Please spell it out and be clear.

Line 164: I am wondering why “using lethal methods” is an objective. This is problematic as I would see lethal methods as one means to an end (controlling cat population). Whether you agree or not there are non-lethal means of managing pests – counting this as an objective seems odd and highlights the problem you have convincing everyone to “be on the same page”. The impression I get as a reader is you want everyone to accept that lethal control is successful and appropriate. Yet you’ve not discussed its efficacy or welfare issues. This makes me very sceptical about your arguments. I would think “the reduction of free roaming cat population numbers” is the objective.

Line 168: Point 4 needs some clarification. The improvement of the welfare of WHICH cats? The free roaming cats who aren’t controlled? Would this be detrimental to native wildlife? Or do you mean welfare of domestic cats who might otherwise interact with free-roaming cats? Are you talking about making control methods more humane? This is very unclear.

Line 171: You mention ‘responsible’ a few times, it may be worth listing that your idea of ‘responsible’ is represented by the contents of the “what” column in your table.

Re registration, this also needs to be updated by owners every time their personal details eg address change.

Line 183: be specific, don’t refer to “the issue” – are you talking about free-roaming cat population control.

Line 193-196: Desexing may be common among owned cats, but if the problem is free-roaming cats, what about semi-owned and ‘feral’ cats?

Line 213: delete “past” (it’s all past)

Line 216: close bracket

Line 219: delete “self”, retain evident

Line 230: Finally you reveal that your concern is with “problematic behaviours such as non-participation in cat management activities”. This seems problematic to me…are you expecting everyone in society to down tools and participate in free-roaming cat management? Or do you mean non-compliance with ‘responsible ownership’ by cat owners = non-participation in cat management activities? Persons who own cats who may be fearful of their cats being caught up in control of free-roaming cats may see their role as a ‘good’ owner in protecting their cat from potential harms and thus abstaining from or even objecting to management. Similarly, persons who don’t agree with methods or means of control may feel they are doing good by abstaining from or objecting to management. Hence I think if you are going to label this as problematic you need to address these issues at length.

Line 282-288: Costs may also be an issue. The Accessibility of Veterinary Care Coalition report talks about a number of barriers to seeking veterinary treatment for cats and dogs.

Table 3: I think it is worth discussing the ethics of these techniques. For example, I think coercing someone into participating in an animal control program has ethical implications. It also has welfare implications, eg if persons are not trained or have negative animals towards animals or the program, animals may be harmed. It may be useful to refer to the work of Hemsworth re how negative and positive attitudes of people working with animals can impact them.

While you mention training re traps they also require monitoring.

I would also query local regulations re fence building. In most places people cannot simply build a fence.

While it is not the focus of the paper I am concerned not much thought has gone into addressing unintended harms of the actions you want people to perform, including and importantly on non-target species.

Line 332-335: How do you avoid that?

Line 338-340: I find this hard to believe. Surely in accordance with everything you have argued so far, targeting the message to the audience is key? Mass media would be unsuitable given the divergence of opinions and different strategies you have outlined?

Line 346: Have you got an example program? Eg the Good Neighbour Project won an award recently for this.

Line 368: I think this should be referenced. Preferred by who? There are textbooks on evidence and scientific method, and papers about study design, that may support this claim.

Line 373: Please strike out or support. Imperfect evidence is DANGEROUS. You have written about myths and misconceptions and how these can be a barrier to GOOD practice. There are papers about interventions eg drugs, surgeries that support their use yet they are taken off the market/stopped when post-marketing data reveals in fact in a bigger, broader population they caused harms.

Line 382: There are ethical and logistic considerations around the measurement of behaviour of humans, as any human ethics committee will tell you.

Line 394: There also needs to be awareness of potential unintended harms, especially when involving laypersons in control programs.

Line 406: The conclusion states “ We have shown how behavioural science approaches can be used to improve free-roaming cat management outcomes by designing more effective interventions to change human behaviour”.  This is not the case. The authors have discussed some literature and identified principles by which they believe human behaviour change may be achieved. They have not tested nor proved their efficacy.

Line 410: I don’t feel this paper defines the issue well.

Author Response

Thank you for your insightful comments.

Reviewer 1:

Line 12: “getting everyone on the same page” seems a bit of a leap and in an issue as divisive as free roaming cat management, highly unlikely despite the most exceptional communication. I wonder if an alternate expression would be better? This is repeated in lines 86-87 and I feel is not transparent. The authors acknowledge strong divergent opinions, yet want everyone on the “same page”, ie agreeing with a particular view. I think the authors are obliged to state their view up front.

We agree with your comment and have endeavoured to be more concise; line 12 – amended to engaging everyone and gaining consensus, line 87 (now line 90) – removed.

Line 22: I would question the goal of maximising the impact of free-roaming cat activities. This is loaded. Isn’t the desired outcome maintenance or increase of biodiversity?

We have rewritten the abstract and this comment is no longer relevant.

Line 35: Would this not also be the case for rabbits?

Yes, but we have said uncommon not unique.

Line 72: It would be useful for the reader to cite these examples so they can read further.

We are conscious of the length of this paper, and as this is not directly related to the paper’s scope we have provided a number of review references which the reader can follow up if they are interested.

Line 73: delete “control”. You could just use methods as killing is the form of population control utilised. There is no reference to whether these methods are humane or not. I think it is important to make this clear.

We have re-written these few sentences to be more concise. It is not in the scope of this paper to discuss the pros and cons of cat control methods. These issues are discussed in the review papers referenced for the reader to follow up if interested.

Lines 77-80: Surely social license is also predicated on the humaneness of killing and other control methods?

Yes humaneness of control methods would probably be an important issue considered in the negotiations and decision-making with stakeholders, particularly around the ethics of managing this species. As you feel this is an important issue we have included it.

Line 83-86: There is also literature documenting that people are more likely to act if they believe their actions will make a difference. This is documented in a range of settings, whether its about using PPE to reduce the spread of equine influenza or implementing changes to reduce anthropogenic climate change.

Yes this belief is a type of motivation. In this sentence we are generally referring to all types of motivations including people’s beliefs, outcome expectations, past experiences, peer-pressure etc.

Line 87: Whenever you state that research has shown something, you are required to cite that research. This paragraph makes strong statements but doesn’t tell us where this info is sourced from.

References have been added.

Line 94: Avoid stating “many theories”. How many? Also in line 99. Give an example, but don’t say “many”.

Now line 119 – it is difficult to give an exact number so we have given an example as suggested. Again for line 99 (now line 124) – it is difficult to give a precise number. This sentence has been re-written to avoid the use of ‘many’.

Line 105-106: What do you mean by “these behaviours” – which ones?

Now line 130 – this has been clarified – these cat management behaviours.

Line 140: Suggest add “we propose” re the common 4 principles

Now line 165 We feel this adds nothing so have not added.

Line 160: Not all of these questions are answered by your points. Please spell it out and be clear.

Now line 189 (note further questions have been added). We have put more thought and detail into our example to make it easier to follow.

Line 164: I am wondering why “using lethal methods” is an objective. This is problematic as I would see lethal methods as one means to an end (controlling cat population). Whether you agree or not there are non-lethal means of managing pests – counting this as an objective seems odd and highlights the problem you have convincing everyone to “be on the same page”. The impression I get as a reader is you want everyone to accept that lethal control is successful and appropriate. Yet you’ve not discussed its efficacy or welfare issues. This makes me very sceptical about your arguments. I would think “the reduction of free roaming cat population numbers” is the objective.

Now line 196 - Yes we agree and the statement has been amended

Line 168: Point 4 needs some clarification. The improvement of the welfare of WHICH cats? The free roaming cats who aren’t controlled? Would this be detrimental to native wildlife? Or do you mean welfare of domestic cats who might otherwise interact with free-roaming cats? Are you talking about making control methods more humane? This is very unclear.

Now point 2, line 195 – this has been clarified.

Line 171: You mention ‘responsible’ a few times, it may be worth listing that your idea of ‘responsible’ is represented by the contents of the “what” column in your table.

With the change in details of the example this is no longer required.

Re registration, this also needs to be updated by owners every time their personal details eg address change.

This has been added to the ‘how often’ column

Line 183: be specific, don’t refer to “the issue” – are you talking about free-roaming cat population control.

Now line 214 – this has been clarified

Line 193-196: Desexing may be common among owned cats, but if the problem is free-roaming cats, what about semi-owned and ‘feral’ cats?

We have stated in line 204 that these Tables are only referring to the owned cat aspect of the example for simplicity.

Line 213: delete “past” (it’s all past)

Now line 266 – done.

Line 216: close bracket

Now line 269 – done.

Line 219: delete “self”, retain evident

Now line 272 – done.

Line 230: Finally you reveal that your concern is with “problematic behaviours such as non-participation in cat management activities”. This seems problematic to me…are you expecting everyone in society to down tools and participate in free-roaming cat management? Or do you mean non-compliance with ‘responsible ownership’ by cat owners = non-participation in cat management activities? Persons who own cats who may be fearful of their cats being caught up in control of free-roaming cats may see their role as a ‘good’ owner in protecting their cat from potential harms and thus abstaining from or even objecting to management. Similarly, persons who don’t agree with methods or means of control may feel they are doing good by abstaining from or objecting to management. Hence I think if you are going to label this as problematic you need to address these issues at length.

The example we gave to illustrate this point was unclear so we have used a more concise example to illustrate what we mean by a desired behaviour.

Line 282-288: Costs may also be an issue. The Accessibility of Veterinary Care Coalition report talks about a number of barriers to seeking veterinary treatment for cats and dogs.

Now line 334 - yes cost may be an issue. Our intention is to provide an example to illustrate the point that not all techniques are well suited to all situations and in this instance we chose access to vet care.

Table 3: I think it is worth discussing the ethics of these techniques. For example, I think coercing someone into participating in an animal control program has ethical implications. It also has welfare implications, eg if persons are not trained or have negative animals towards animals or the program, animals may be harmed. It may be useful to refer to the work of Hemsworth re how negative and positive attitudes of people working with animals can impact them.

This purpose of this Table is purely to illustrate that different techniques are used for different behavioural factors. There is not room to discuss the ethics of every techniques. We feel that the ethical implications of this work has generally been covered with the addition of the APEASE component inserted in the text just after Table 3.

While you mention training re traps they also require monitoring.

How often they should be monitored would be part of how to use then proficiently, along with where to set them, what bait types to use etc

I would also query local regulations re fence building. In most places people cannot simply build a fence.

Yes that may also be a factor limiting people’s opportunity to do this behaviour, but we have focussed just on the training aspect to illustrate its use when the barrier is a ‘capability’ factor.

While it is not the focus of the paper I am concerned not much thought has gone into addressing unintended harms of the actions you want people to perform, including and importantly on non-target species.

We feel we have addressed this with the addition of the APEASE component inserted in the text just after Table 3.

Line 332-335: How do you avoid that?

Now line 388 – We have removed these other points as they were not relevant to the current discussion

Line 338-340: I find this hard to believe. Surely in accordance with everything you have argued so far, targeting the message to the audience is key? Mass media would be unsuitable given the divergence of opinions and different strategies you have outlined?

Now line 389 – yes targeting the message is important, but you may also recall that behaviour change is a process in itself – so if the identified barrier is initially awareness, and the targeted group uses a mass media channel, the first step might be to use this channel. Once awareness has increased then you progress to the next stage of adoption of a behaviour.

Line 346: Have you got an example program? Eg the Good Neighbour Project won an award recently for this.

Now line 399 – thank you for this example, we have included it in the paper.

Line 368: I think this should be referenced. Preferred by who? There are textbooks on evidence and scientific method, and papers about study design that may support this claim.

Now line 421 - a reference has been added.

Line 373: Please strike out or support. Imperfect evidence is DANGEROUS. You have written about myths and misconceptions and how these can be a barrier to GOOD practice. There are papers about interventions eg drugs, surgeries that support their use yet they are taken off the market/stopped when post-marketing data reveals in fact in a bigger, broader population they caused harms.

Now line 425, we have worded this sentence badly, the evidence from quasi-experiments is not imperfect, just not as strong as random experiments. We have removed as suggested.

Line 382: There are ethical and logistic considerations around the measurement of behaviour of humans, as any human ethics committee will tell you.

Now line 438 – Yes we appreciate these considerations, that is why we have added the statement about how measuring it directly is not easy.

Line 394: There also needs to be awareness of potential unintended harms, especially when involving laypersons in control programs.

Now line 441 – this has been addressed in the intervention design discussion – see from line 351.

Line 406: The conclusion states “ We have shown how behavioural science approaches can be used to improve free-roaming cat management outcomes by designing more effective interventions to change human behaviour”.  This is not the case. The authors have discussed some literature and identified principles by which they believe human behaviour change may be achieved. They have not tested nor proved their efficacy.

Now line 457 – we disagree, the methods we presented are not just what ‘we believe’ but what has been proven to work in the behavioural science literature in other fields such as health, and environmental management. We have slightly reworded this sentence to indicate this point.

Line 410: I don’t feel this paper defines the issue well.

Now line 464 – The intention of this paper is not to define any particular issue but to illustrate why practitioners should define their issue first and understand the relevant behaviours before thinking about an intervention strategy.

Reviewer 2 Report

Whether free-roaming and feral, or much-loved pets, domestic cats are difficult to manage owing to their shy, solitary and usually elusive nature. However, as the authors of this manuscript make clear, no attempts at management will be successful if people are not on board, and this can be a much more serious challenge than managing cats themselves. This article makes an excellent contribution in selectively outlining relevant theories of behaviour, and behaviour change, and then providing a framework that allows more efficient and cost-effective interventions to be designed. It should provide a useful platform for future management of free-ranging cats to be implemented. The overview of the relevant literature is well focused, and the resulting framework is derived logically from the key findings of the overview. The writing is clear. Overall, I found this to be an insightful and very interesting manuscript, and found just two very minor issues to note.

1) Line 66: The Commonwealth Environment Protection and Diversity Act 1999 should be the Environment Protection and Biodiversity Conservation Act 1999, and 'Commonwealth' should not be italicised as it is not part of the name of the Act.

2) Line 361: I think some rewording is needed here. Randomisation is certainly a key part of experimental design, but it isn't intended to ensure that groups are as similar as possible before delivering an intervention. Rather, it is used to ensure that groups, or subjects within groups, are allocated without conscious or unconscious bias. If there are few groups to be randomised, a strictly randomised process may even result in considerable differences between groups. In this situation, interspersion or some other quasi-random approach may be preferred.    

Author Response

Whether free-roaming and feral, or much-loved pets, domestic cats are difficult to manage owing to their shy, solitary and usually elusive nature. However, as the authors of this manuscript make clear, no attempts at management will be successful if people are not on board, and this can be a much more serious challenge than managing cats themselves. This article makes an excellent contribution in selectively outlining relevant theories of behaviour, and behaviour change, and then providing a framework that allows more efficient and cost-effective interventions to be designed. It should provide a useful platform for future management of free-ranging cats to be implemented. The overview of the relevant literature is well focused, and the resulting framework is derived logically from the key findings of the overview. The writing is clear. Overall, I found this to be an insightful and very interesting manuscript, and found just two very minor issues to note.

Thank you for your comments.

Reviewer 2:  

1) Line 66: The Commonwealth Environment Protection and Diversity Act 1999 should be the Environment Protection and Biodiversity Conservation Act 1999, and 'Commonwealth' should not be italicised as it is not part of the name of the Act.

This has been corrected.

2) Line 361: I think some rewording is needed here. Randomisation is certainly a key part of experimental design, but it isn't intended to ensure that groups are as similar as possible before delivering an intervention. Rather, it is used to ensure that groups, or subjects within groups, are allocated without conscious or unconscious bias. If there are few groups to be randomised, a strictly randomised process may even result in considerable differences between groups. In this situation, interspersion or some other quasi-random approach may be preferred.  

This paragraph has been rewritten.

Reviewer 3 Report

This paper describes the adaptation of theories of human behaviour to the issue of cat management, as required to reduce the threat posed by these animals to wildlife populations. I find this paper fascinating and extremely well-written. I am not a psychologist by training but the authors clear explanations with useful examples allowed me (and hopefully others like me) to understand and be engaged with the concepts. It provided a great introduction to the theories considered and then cleverly applied them to the issue at hand. It made me think more about my own area of research and the application of behavioural theory.  All in all, a novel, stimulating and well-considered piece. 

As a last specific point:

L 40- needs an 'or' I think.

Author Response

This paper describes the adaptation of theories of human behaviour to the issue of cat management, as required to reduce the threat posed by these animals to wildlife populations. I find this paper fascinating and extremely well-written. I am not a psychologist by training but the authors clear explanations with useful examples allowed me (and hopefully others like me) to understand and be engaged with the concepts. It provided a great introduction to the theories considered and then cleverly applied them to the issue at hand. It made me think more about my own area of research and the application of behavioural theory.  All in all, a novel, stimulating and well-considered piece. 

Thank you for your comments.

As a last specific point:

L 40- needs an 'or' I think.

This has been added

Round 2

Reviewer 1 Report

The authors have responded to my comments constructively and thoughtfully and I think the paper is much improved.